# Multisite, mixed methods study to validate 10 maternal health system and policy indicators in Argentina, Ghana and India: a research protocol

R Rima Jolivet ![ORCID],[1] Jewel Gausman ![ORCID],[2] Richard Adanu,[3] Delia Bandoh,[4] Maria Belizan,[5] Mabel Berrueta,[5] Suchandrima Chakraborty,[6] Ernest Kenu,[4] Nizamuddin Khan,[6] Magdalene Odikro,[7] Veronica Pingray,[5] Sowmya Ramesh,[6] Niranjan Saggurti,[6] Paula Vázquez ![ORCID],[5] Ana Langer[8]

For numbered affiliations see end of article.

**Correspondence to**
Dr R Rima Jolivet;
rjolivet@hsph.harvard.edu

## ABSTRACT

**Introduction** Most efforts to assess maternal health indicator validity focus on measures of service coverage. Fewer measures focus on the upstream enabling environment, and such measures are typically not research validated. Thus, methods for validating system and policy-level indicators are not well described. This protocol describes original multicountry research to be conducted in Argentina, Ghana and India, to validate 10 indicators from the monitoring framework for the 'Strategies toward Ending Preventable Maternal Mortality' (EPMM). The overall aim is to improve capacity to drive and track progress towards achieving the priority recommendations in the EPMM strategies. This work is expected to contribute new knowledge on validation methodology and reveal important information about the indicators under study and the phenomena they target for monitoring. Validating the indicators in three diverse settings will explore the external validity of results.

**Methods and analysis** This observational study explores the validity of 10 indicators from the EPMM monitoring framework via seven discrete validation exercises that will use mixed methods: (1) cross-sectional review of policy data, (2) retrospective review of facility-level patient and administrative data and (3) collection of primary quantitative and qualitative cross-sectional data from health service providers and clients. There is a specific methodological approach and analytic plan for each indicator, directed by unique, relevant validation research questions.

**Ethics and dissemination** The protocol was approved by the Office of Human Research Administration at Harvard University in November 2019. Individual study sites received approval via local institutional review boards by January 2020 except La Pampa, Argentina, approved June 2020. Our dissemination plan enables unrestricted access and reuse of all published research, including data sets. We expect to publish at least one peer-reviewed publication per validation exercise. We will disseminate results at conferences and engage local stakeholders in dissemination activities in each study country.

## Strengths and limitations of this study

► This research uses innovative methodological approaches to validate indicators for monitoring maternal health policy and maternal health system effectiveness, which are seldom systematically research validated.

► The study scale addresses 10/25 of the metrics from the comprehensive monitoring framework for the 'Strategies toward Ending Preventable Maternal Mortality' designed to monitor distal determinants of maternal mortality that comprise an enabling environment for maternal health and survival.

► The study methods target the underlying constructs that the 10 discrete indicators are intended to measure and provide evidence to validate how well they reflect the phenomena they target for monitoring.

► Systematic sampling across 12 districts in three diverse settings increases the external validity of the results.

► The research does not reflect comprehensive national data but rather is limited to four subnational study settings in each country.

## INTRODUCTION

Sustainable Development Goal (SDG) 3.1.1. targets a global maternal mortality ratio (MMR) of <70 maternal deaths per 100 000 live births by 2030. There were 295 000 maternal deaths in 2017, a global MMR of 211/100 000. If the average annual rate of reduction does not accelerate above 2.9%, the rate from 2000 to 2017, we will miss the target by 1 million preventable maternal deaths worldwide.[1] As countries move through the obstetric transition[2] and maternal deaths shift from direct obstetric to indirect causes, addressing upstream factors is critical to ending preventable maternal mortality. Graham *et al*[3] illustrated the widening range of causes of death between and within countries. Thus,

| **Table 5** | EPMM 11 Key Themes |
|---|---|
| Guiding principles | 1. Empower women, girls, families and communities |
| | 2. Integrate maternal and newborn health, protect and support the mother-baby dyad |
| | 3. Prioritise country ownership, leadership, and supportive legal, regulatory and financial frameworks |
| | 4. Apply a human-rights framework to ensure that high-quality reproductive, maternal and newborn healthcare is available, accessible and acceptable to all who need it |
| Cross-cutting actions | 5. Improve metrics, measurement systems and data quality |
| | 6. Prioritise adequate resources and effective healthcare financing |
| Five strategic objectives | 7. Address inequities in access to and quality of sexual, reproductive, maternal and newborn healthcare |
| | 8. Ensure universal health coverage for comprehensive sexual, reproductive, maternal and newborn healthcare |
| | 9. Address all causes of maternal mortality, reproductive and maternal morbidities and related disabilities |
| | 10. Strengthen health systems to respond to the needs and priorities of women and girls |
| | 11. Ensure accountability in order to improve quality of care and equity |

(table 5) Jolivet RR, Moran AC, O'Connor M, Chou D, Bhardwaj N, Newby H, Requejo J, Schaaf M, Say L, Langer A. Ending preventable maternal mortality: phase II of a multi-step process to develop a monitoring framework, 2016–2030. BMC pregnancy and childbirth. 2018 Dec;18(1):1-3.
EPMM, Ending Preventable Maternal Mortality.

recognition is growing of the importance of social, political, economic and structural factors that impact causes of death and health system responses to them. These include the status of women in societies, the functionality of health systems, access to universal health coverage and reproductive justice, the capacity to register all births and to count all deaths and track their causes and to address all causes effectively. With acknowledgement of the significance of such distal determinants, improving metrics, data quality and measurement capacity to monitor them has taken on greater urgency.

In 2015, the WHO released the 'Strategies toward Ending Preventable Maternal Mortality (EPMM)' (EPMM Strategies),[4] a global guidance document outlining targets and strategies for reducing maternal mortality in the SDG period. Developed through extensive stakeholder consultations, the strategies address the broad spectrum of determinants of maternal health and survival, exemplified in 11 Key Themes.

In 2016, over 150 technical, policy and country experts from 78 organisations worldwide participated in a five-round modified Delphi process to develop a comprehensive monitoring framework for the EPMM Strategies, comprising indicators centred on its 11 Key Themes. A set of 25 indicators, plus six indicator stratification factors to allow tracking of inequities and data transparency, were identified by participants as the strongest available measures for tracking progress towards the priority recommendations in the report.[5] The organising framework of the EPMM 11 Key Themes and menu of associated indicators were designed to support national decision-makers in identifying priority areas for improvement in their context, and in tracking and driving improvement in those areas deemed of greatest relevance and urgency.

Most efforts to assess maternal health indicator validity focus on measures of service coverage[6–9] and, to a lesser extent, quality and reliability of service delivery.[10 11 12] Fewer measures overall focus on the upstream enabling environment for maternal healthcare provision, and they are typically not subjected to validation research.[13] Methods for validation of health system and policy-level indicators are, therefore, not well described.

In 2019, the WHO "Mother and Newborn Information for Tracking Outcomes and Results" (MoNITOR) expert working group commissioned a landscape analysis based on interviews with experts in maternal and newborn health (MNH) measurement to better understand how they conceptualize indicator validity, approaches to validation, and gaps in the science[11]. The analysis identified gaps in research on indicator validity conducted in low- and middle-income (LMIC) settings and poor knowledge translation about indicator validity to those settings. As a result, it found little application of information on validity in the evaluation and selection of indicators for national and subnational monitoring. Some types of indicators, in particular, lacked research-based validation, for example, those for monitoring women's satisfaction and experiences of care; abortion services as well as indicators derived from facility and routine data systems and the policy environment. Recommendations included engaging national stakeholders in discussions and research on indicator validity, and focusing beyond diagnostic-style, criterion-related validity to encompass the meaningfulness of indicators, including the accurate definition of their underlying constructs and their utility to drive improvement.

Benova *et al*[14] published a conceptual framework compiling definitions of indicator validity (table 1) and approaches for assessing its various dimensions, based on interviews with practitioners of MNH measurement. The framework includes methodological approaches for assessing validity of indicators for tracking health policy

| Table 1 | What is indicator validity? |
| --- | --- |
| **What does indicator validity mean?**[14 20] | |
| **Validity** asks, 'Is this measurement truly representative of the concept under study?' | |
| **Selected types of validity** | **Definition** |
| Content validity | Does the indicator fully represent the content domain or concept to be measured? |
| Criterion-related validity | How does the value of an indicator compare to an objective measure of truth? |
| Construct validity | Do two indicators that are purported to measure the same construct 'behave' in the same way? |

and health system factors and calls for more research in this domain. We used this framework to inform the development of our research methods, based on specific validation questions of relevance to each indicator undergoing assessment.

To fill critical gaps in the assessment of maternal health measure validity, the present protocol describes multicountry research to be conducted in Argentina, Ghana and India at both national and subnational levels. The overall aim of the study is to improve maternal health measurement by validating 10 indicators from the EPMM monitoring framework, in order to drive improvement and track progress towards achieving the priority recommendations outlined in the EPMM Strategies. Of note, this research assesses 40% of the indicators in the set of EPMM metrics designed to allow countries and global partners to monitor critical dimensions of the upstream enabling environment for maternal health. Furthermore, the indicators validated through this research reflect a broad range of these distal determinants, as they correspond to 9 out of the 11 EPMM Key Themes (figure 1).

## METHODS AND ANALYSIS

This observational study explores the validity of 10 indicators from the EPMM monitoring framework. It uses mixed methods, including (1) cross-sectional review of secondary policy, legal and regulatory data, (2) retrospective review of facility-level patient and administrative data and (3) collection of primary, quantitative, cross-sectional data from health service providers and clients. Standard approaches for assessing the validity of policy and health system indicators are not available; therefore, we developed a specific methodological approach to validate each indicator, tailored to test the validation questions that reflect the specific aims and research questions relevant to each indicator undergoing validation and its underlying construct. Because there is no standard approach (metric or framework) for assessing validity of indicators of upstream health system functionality, we have developed a tailored analytical plan with appropriate statistics to compare the values of the reported indicators to evidence collected in each case. In two specific cases, two indicators designed to monitor a similar construct are compared with each other to explore their convergence and whether indicator adjustment could improve

measure validity for that construct. These two indicator pairs share the same validation research questions and are studied in tandem. Thus, the validity of the 10 EPMM indicators is evaluated via seven separate assessments, or validation exercises.

The 10 EPMM indicators under study and the specific validation research questions for each indicator appear in table 2. Nine indicators will be validated in all countries, and one additional indicator is to be validated in Ghana only due to local interest. Data collection began in January 2020 was suspended due to COVID-19, resumed May 2020, and is expected to be completed by November 2021 in all settings.

## Research settings

The research will be coordinated by a multicountry team of partners from all three countries and the USA. Country partners were selected through a competitive process based on proposal strength and geographic diversity. One application was selected from Africa, Asia and Latin America/Caribbean, respectively, based on World Bank classification.[15]

The research will comprise national and subnational data; however, fieldwork will be conducted in subnational settings in each country. Four districts/provinces in each country were selected for primary data collection. Sites were selected through a purposive, two-staged sampling approach based on a composite index of key maternal health indicators reflecting antepartum, intrapartum and postpartum care coverage and MMR, used as a proxy of health system performance. First, one state/region in the highest-performing quartile of the index and one state/region in the lowest-performing quartile were selected. Second, one highest-performing district/province and one lowest-performing district/province were selected within each state/region. In Argentina, some adjustments to the standard site selection protocol were implemented. Due to low population density, terciles were used. In addition, because there was almost no geographic variability in skilled birth attendance and early postnatal care coverage in data from Argentina where most births take place in facilities, Uterotonic Administration at Birth was substituted in the index for this country. Finally, to avoid over-representation of data from Buenos Aires province due to its disproportionate size (total population of over 16.5 million), Region V of the province was selected in

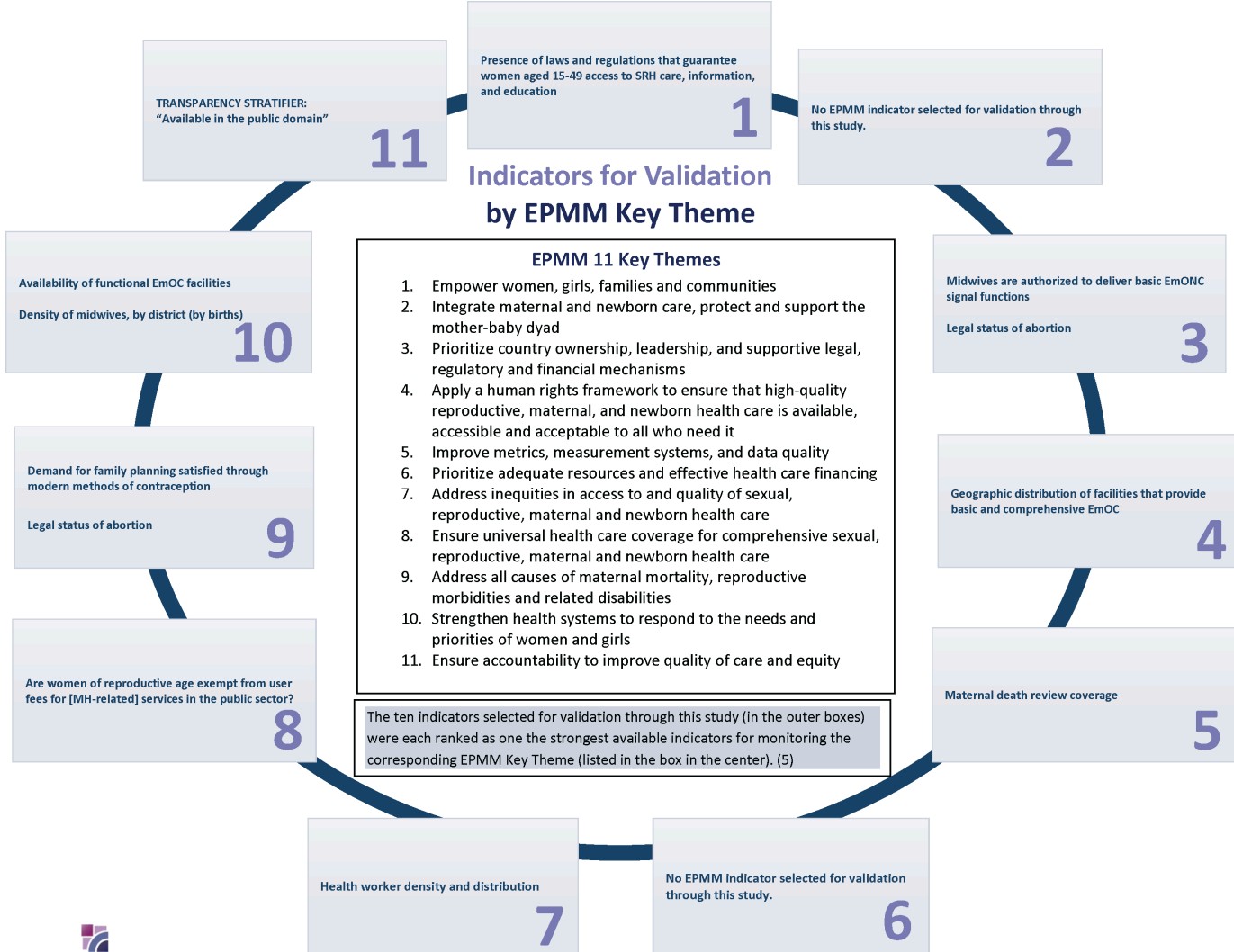

**Figure 1** Ten indicators for validation and their corresponding EPMM key themes. EPMM, Ending Preventable Maternal Mortality. SRH; Sexual Reproductive Health, EmOC;Emergency Obstetric Care, EmONC ; Emergency Obstetric and Neonatal Care

consultation with the National Ministry of Health to represent the province. Region V of Buenos Aires province comprises 13 counties, a total population of 3 432 962, 16 hospitals and 319 primary health centres and reflects similar sociodemographic, geographic and health system characteristics as the entire province in table 3.

### Data sources, participants and sampling

Data required for validation vary by indicator; details of the data sources, participants and sampling for each indicator are presented in table 4.

In general, three types of data will be collected: policy/administrative, facility, and individual data.

### Policy/administrative data

We will systematically search for national and subnational policies, laws and regulations through a comprehensive desk review of relevant source documents in each country. Country research teams will consult with subject matter experts and data custodians to ensure that all relevant documents were captured. Country-specific data will

also be collected from global databases and repositories, as required by each indicator. Furthermore, administrative and patient-level data will be collected from district/provincial-level health management information systems (HMIS).

### Facility data

Facilities will be selected based on data requirements for each indicator, using a multistage sampling plan (figure 2). In the first stage, we will conduct a census of all public and private registered health facilities in each study district/province. For some indicators, data will be collected from all facilities in the census. Next, we will determine which maternal health-related services are provided at each facility in the census. We will collect information on provision of services within the five categories in the WHO Maternal Newborn Child and Adolescent Health (MNCAH) Policy Survey: (1) caesarean section, childbirth (normal delivery), delivery-related pharmaceutical products and medical supplies, (2) family

**Table 2** Indicators for validation and validation questions

| Indicators for validation | Validation questions |
|---|---|
| 1. Legal status of abortion | 1. How does the law, as expressed in the national statute, compare to the countdown indicator metadata and to the information available on the WHO Global Abortion Policies Project Database for the country? (Criterion validity)<br>2. Is there evidence that providers are consistently applying the law for each of the grounds on which abortion is legal? (Construct validity) |
| 2. If fees exist for health services in the public sector, are women of reproductive age (15-49) exempt from user fees for (maternal health -related) services | 1. Does the free care law or policy in the country provide all of the categories of services included in the indicator free of charges or fees to users? (Criterion validity)<br>2. For the categories of services that should be free according to the law/policy in the country, is there evidence that women are paying user fees for them? (Construct validity)<br>3. If evidence is found that demonstrates that women are paying for services that are supposed to be free according to the law/policy in the country, is there evidence that user fees are being levied in a systematically differential way to women? (Equity analysis) |
| 3. Health worker density and distribution (per 1000 population)<br>4. Density of midwives, by district (by births)<br>(*The validity of these two indicators designed to measure a related construct will be evaluated in tandem using the same research validation questions.) | 1. How does the definition of a midwife/midwifery professional on record in the country compare to the ILO definition and to the ICM midwifery competencies? (Criterion validity)<br>2. What proportion of practising midwives meet the ICM standard for competency as evidenced by an analysis of the tasks they have performed in the last 90-day period? (Construct validity)<br>3. How does the value of the estimate differ based on the denominator used? (Convergent validity) |
| 5. Midwives are authorised to deliver BEmONC | 1. Does the national regulatory framework in country that authorises midwives/MPs to deliver BEmONC match was has been reported for this indicator for all seven signal functions? (Criterion validity)<br>2. For signal functions that midwives/MPs are authorised to perform according to national regulations, is there evidence they have performed these tasks in settings where EmONC is provided in last year? (Construct validity) |
| 6. Availability of functional EmOC facilities<br>7. Geographic distribution of facilities that provide basic and EmOC<br>(*The validity of these two indicators designed to measure a related construct will be evaluated in tandem using the same research validation questions.) | 1. Is there evidence from facilities designated as B/CEmONC to demonstrate that they have performed all seven signal functions in last 3 months as defined in the metadata for these indicators? (Construct validity)<br>2. How does the value of the indicator differ based on the denominator used: 500 000 population/district vs 20 000 birth/district vs travel time (<2 hours for BEmONC)? (Convergent validity) |
| 8. Maternal death review coverage | 1. How does evidence from the facility level on maternal death reviews compare to the coverage of maternal death reviews reported at district level, through state or district reporting programmes? (Criterion validity)<br>2. How does the number of facility deaths captured through review of facility patient register data compare to the number of deaths reported at the district level? (Convergent validity)<br>3. How does the value of the indicator reported compare to the value calculated using primary data? (Convergent validity) |
| 9. Demand for family planning satisfied through modern methods of contraception | 1. How does a direct measure of demand satisfaction for family planning (woman's self-report) compare to the assigned result provided by the DHS algorithm derived from the responses to the series of questions used to calculate the indicator (same woman surveyed) (Construct validity)?<br>2. How does the value of the indicator vary based on a new data source/estimation method compared with an established source/method? (Convergent validity) |

Continued

| Table 2 | Continued | |
| --- | --- |
| **Indicators for validation** | **Validation questions** |
| 10. Presence of laws and regulations that guarantee women aged 15–49 access to sexual and reproductive healthcare, information and education (*Assessment of the validity of this indicator will be conducted using data from Ghana only due to local stakeholder interest.) | 1. Do the laws or regulations as recorded on the national statute in Ghana match the definition of the indicator, fully including all 13 components? (Presence of laws) (Criterion validity)<br>2. How does the value of the indicator change using two different methods of computation (scoring)? (Convergent validity) |

B/CEmONC, Basic and Comprehensive Emergency Obstetric and Neonatal Care; BEmONC, basic emergency obstetric and neonatal care; EmOC, Emergency Obstetric Care; EmONC, Emergency Obstetric and Neonatal Care; ICM, International Confederation of Midwives; ILO, International Labour Organization; MPs, Midwifery Professionals.

planning, (3) antenatal care and insecticide-treated bed nets, (4) postnatal care for mother, (5) testing and treatment for sexually transmitted infectious diseases and cervical cancer screening.[16] Although infertility management is included in the WHO MNCAH Policy Survey, it is not in our study.

Thereafter, we will replicate the methodology used in Demographic and Health Surveys (DHS)[17] to define primary sampling units (PSUs), which are typically census tracts or discrete villages, depending on the country. We will randomly select 20 PSUs in each study district/province based on probability proportionate to size. Finally, we will define eligible facilities for each indicator within the sampled PSUs based on the services they provide relevant to the specific validation questions for that indicator. Eligible facilities for each indicator will include all lower level primary health facilities within the PSUs that provide the relevant maternal health-related services, plus all higher level facilities across the district/province.

| Table 3 | National and subnational research settings | |
| --- | --- | --- |
| **Country** | **State/region** | **District/province** |
| Argentina | Centro | Buenos Aires Region V |
| | | La Pampa |
| | Noroeste | Salta |
| | | Jujuy |
| Ghana | Brong Ahafo | Techiman North |
| | | Sunyani Municipal |
| | Northern | Bunkpurugu-Yunyoo |
| | | Tolon |
| India | Tamil Nadu | Thiruvallur |
| | | Krishnagiri |
| | Uttar Pradesh | Meerut |
| | | Gonda |

### Individual data

Within study districts/provinces, we will collect primary, quantitative, individual-level data from study participants via surveys conducted at facilities and in communities. Eligible facility-based participants will include administrators; maternity care clinicians (midwives/midwifery professionals and clinical cadres legally authorised to provide induced abortions); women who received an included maternal-health related service at an eligible facility and their chosen companions if they had a complicated childbirth or caesarean birth. Within eligible facilities, we will obtain a sample of staff participants as detailed in table 4. We will enrol 1040 women of reproductive age who received maternal health services in each country, representing 20 women per service/district for 260 women total per district.

Eligible community-based participants will include women of reproductive age (15–49 years). We will use the same 20 PSUs to obtain the community-based sample of women. Within each, a house listing exercise will identify households with women of reproductive age (15–49 years). From this list, 18 households per PSU will be randomly selected and 1420 women will be recruited, based on the following sample size calculation: $n = \frac{Z^2 * pq}{d^2}$, where Z is the standard normal deviate, p is the proportion of population with characteristic, q is the proportion of population without characteristic, d is the degree of accuracy required. The sample size derived through this calculation (n=96) was further adjusted to reflect an estimated 10% non-response rate, a design effect of 2 to account for clustering and a multiplier of 1.68 to account for the low prevalence of modern contraception in each country, yielding a final sample size of 355 women per district/province. Household surveys are infeasible in Argentina due to low population density, vast distances between households and lack of cultural acceptance. Therefore, interviews will be conducted with a random sample of 360 women per district exiting from eligible facilities.

**Table 4** Participants and sampling plan detailed by validation exercise

| Validation exercise | National/subnational data sources | Facility-Level data | | | Individual-Level data | | |
|---|---|---|---|---|---|---|---|
| | | Facility selection | Facility sampling plan | Data source | Participant selection | Participant sampling plan | Data source |
| 1 | National/subnational document review Countdown 2030 country profile WHO Global Abortion Policies Project Database | Sample of facilities within 20 PSUs All higher-level facilities in study districts/provinces | All facilities that perform at least one maternal health-related service | No facility-level data collected | All health service providers who belong to professional cadres that are legally authorised to provide abortion within the study setting | All eligible health service providers in all eligible facilities | Survey administered to eligible providers |
| 2 | National/subnational document review WHO Maternal Newborn Child and Adolescent Health Policy Survey | Sample of facilities within 20 PSUs All higher-level facilities in study districts/provinces | All facilities that perform at least one maternal health related service | No facility-level data collected | Chief financial officer (or similar administrative position) for each facility Woman who received maternal health-related services Companion of choice (eg, family member or friend, if applicable) for women who had a complicated birth and/or underwent a caesarean | All chief financial officers in all eligible facilities All eligible women (or their companion of choice) leaving eligible facilities | Interviews with chief financial officers Exit interviews with women or their companion of choice |
| 3a | National/subnational document review District/provincial demographic data including total population, number of women of reproductive age, number of births, and number of pregnancies | Census of all facilities in study districts/provinces | All facilities that perform at least one maternal-health related service | Facility staff listing | All currently employed professionals who meet the International Labour Organization's description of midwifery professionals or midwifery associate professionals | All eligible providers in all eligible facilities (in facilities with more than 50 eligible providers, a random sample of 50 providers will be drawn). | Survey administered to midwifery professional/midwifery associate professionals |
| 3b | National/subnational document review | Sample of Facilities within 20 PSUs All higher-level facilities in study districts/provinces | All B/CEmONC facilities | Not applicable | All currently employed professionals who meet the International Labour Organization's description of midwifery professionals or midwifery associate professionals | All eligible providers in all eligible facilities (in facilities with more than 50 eligible providers, a random sample of 50 providers will be drawn). | Survey administered to midwifery professional/midwifery associate professionals |

Continued

**Table 4** Continued

| Validation exercise | National/subnational data sources | Facility-Level data | | | Individual-Level data | | |
| --- | --- | --- | --- | --- | --- | --- | --- |
| | | Facility selection | Facility sampling plan | Data source | Participant selection | Participant sampling plan | Data source |
| 4 | Districtprovincial demographic data including total population, number of women of reproductive age, number of births and number of pregnancies | Census of all facilities in study districts/ provinces | All facilities that provide birth care in each district/ province | Facility GIS locational data | Not applicable | Not applicable | Not applicable |
| 5 | Health Information System Data Death Reviews reported to district/ province | Census of all facilities in study districts/ provinces | All facilities that provide birth care in each district/ province | Administrative data registers | Not applicable | Not applicable | Not applicable |
| 6 | Not applicable | Not applicable | Not applicable | Not applicable | Community-based sample of women* | Women aged between 15 and 49 years in study districts | Individual interview |
| 7 | National/subnational document review United Nations 12th Inquiry Among Governments on Population and Development, Module II (Fertility, Family Planning, and Reproductive Health) Survey | Not applicable | Not applicable | Not applicable | Not applicable | Not applicable | Not applicable |

B/CEmONC, Basic and Comprehensive Emergency Obstetric and Neonatal Care; GIS, Goegraphic Information System; PSU, primary sampling unit.

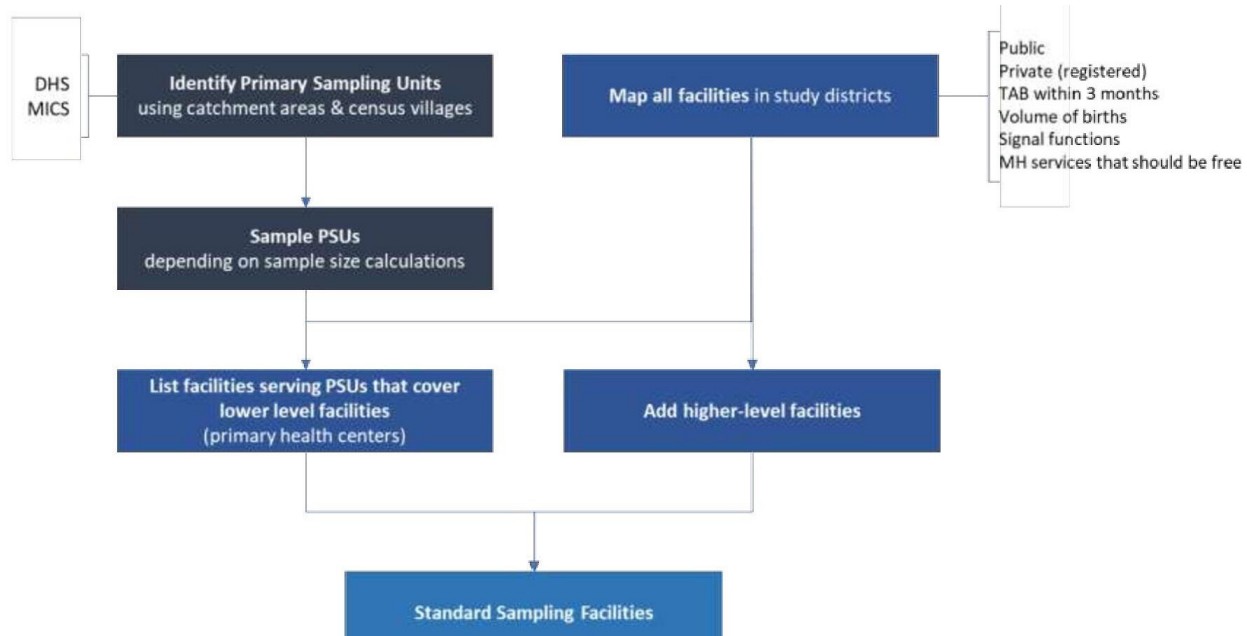

**Figure 2** Schematic of standard sampling plan for facilities. DHS; Demographic and Health Surveys, PSU; primary sampling unit, MICS; Multiple Indicator Cluster Surveys, TAB;therapeutically induced Abortion, MH; maternal health.

## Eligibility and exclusion criteria

Facility eligibility criteria are detailed above. Participants will be considered eligible if they belong to one of the targeted participant groups listed above, and/or have received an included maternal health-related service and meet the age of majority to consent or else provide assent along with parental consent if younger (less than 18 years old in Ghana and India; less than 16 years old in Argentina).

Exclusion criteria include not being proficient in the local language; not meeting the age of majority in the country, district or province unless they can provide parental consent; being unable, unwilling or lacking capacity to provide consent or assent.

## Public and patient involvement

No patients were involved in the design, conducting, reporting or dissemination of this study. We will engage local country stakeholders in a dissemination activity in each study country. We will disseminate results to district/provincial government units and participating health facilities as appropriate, to ensure that they can be used to drive progress and improvement in the study settings.

In the following section, we describe in detail the specific methodology and analytical plan for each indicator.

## Indicator number 1: validating 'legal status of abortion' as an indicator of equal access under the law
### ims

(1) To verify that the 'legal status of abortion' indicator reported globally by each country accurately reflects the laws and statutes on record; and (2) to look for variation at the provider and facility level of the application of the legal categories under which abortion is lawful (legal grounds) and, thus, the accessibility of induced abortion.

### Methods

This validation exercise will use mixed methods exploring two validation questions to test the global indicator on legal status of abortion. We will conduct a desk review of the legal grounds for induced abortion expressed in national laws (subnational laws, in Argentina), also capturing any requirements for eligibility on each legal ground articulated in the legal statutes. We will conduct surveys with health professionals whose scope of practice authorises them to provide abortion services in each setting to explore provider knowledge of the legal grounds for abortion in their jurisdiction and provider practices for determining patient eligibility on each legal ground, providing abortion services or referrals.

For the first validation question, we will compare and describe any differences between legal statutes in each country, reported data in the Countdown indicator, and the WHO GAPP database. For the second, we will tabulate the number of accurate survey responses among abortion providers on the legal grounds for abortion in their jurisdiction. We will explore any variance in provider requirements to access abortion for each legal ground in the country to look for differences in the application of the law across providers and facilities. Descriptive statistics will be reported and we will stratify the results to look for systematic variance.

## Indicator number 2: validating reported policies on free maternal health-related services in the public sector

### Aim

To verify that no charges, formal or informal, are assessed for services included in the indicator that are supposed to be free by law and to describe variance between the law and primary data sources.

### Methods

We will conduct a desk review of national and subnational laws and policies on free care provision. We will administer surveys to chief financial officers (or similar administrative position) within participating health facilities to collect data on formal fees or payments charged for any included services and the rationale. We will conduct interviews with women exiting eligible facilities to ask about formal and informal charges for any services received. If a woman had a complicated birth or caesarean section and a companion of choice (eg, family member or friend) is present who was at the facility during the birth, we will interview them as well about any charges they may have paid on her behalf.

We will use comparative analysis to detect and describe differences between service categories designated as free to users in the national statutes, and the most recent data reported by the country in the WHO MNCAH Policy Survey. We will estimate the per cent of women paying fees for each type of service. Universal applicability of the policy implies that 0% of women pay fees for maternal health services in the public sector. We will test the significance in the difference using a one-sample test of proportion. We will use a $\chi^2$ test to determine whether fees are levied in a systematically different way to various types of women using the EPMM standard equity stratifiers. Results will be reported by service type and client demographics, and the value of the indicator expressed each way will be compared with explore optimal construct validity.

## Indicators 3, 4 and 5: validating critical measures for monitoring adequacy of the midwifery workforce

### Aim

To strengthen measurement of midwifery workforce adequacy. Three aspects of adequacy are reflected: density (number to meet need), distribution (accessibility) and both competency and authorisation to provide essential care (availability).

Two nested validation exercises are included. The aims of the first one are: (1) to compare midwifery professionals' scope of practice in each country to international reference standards from the International Labour Organization's (ILO) definitions for midwifery professionals and associate professionals and to the International Confederation of Midwives (ICM) Essential Competencies for Midwifery Practice and (2) to compare estimates derived from two indicators to measure the same construct

(density and distribution of midwives), to explore consistency (convergent validity), evidence that one measure is more accurate or a more efficient way to capture the construct and whether adjusting the numerator and/or denominator provides a better estimate.

The second validation exercise aims to verify whether midwives and midwifery professionals are authorised to perform basic obstetric and neonatal care (BEmONC) functions and whether they do so in practice.

### Methods

We will conduct document review to compare the national scope of practice for midwifery professionals on record in each country to the ILO and ICM descriptions for midwifery personnel. We will review national laws and regulations that authorise midwifery professionals' scope of practice in each country to verify what is reported by the country in the MNCAH Policy Survey. Then, we will recruit a representative sample of midwifery professionals employed within all participating facilities providing maternal health-related services in each study district. We will administer a survey asking respondents whether they have the skills necessary to perform each competency and/or BEmONC signal function; how they obtained those skills; the frequency and recency of behaviours related to each competency or reasons for non-performance of these behaviours in their current job.

We will report the percent agreement between the national scope of midwifery practice and the ILO tasks, the ICM competencies and the variance between them. We will calculate the percent (%) of midwives whose current practice meets the international standard reflected in the ICM competencies as well as the average competency of midwives in the sample, stratified by facility type (public, private) and geography (urban, rural). Last, we will compare the value of the indicator for density and distribution of midwives, adjusted using different numerators and denominators. For numerators, we will calculate the value using the number of midwives on facility rosters, those who meet the ILO definition, and those who meet the ICM competencies. For the denominator, we will examine the value of the indicator using different population parameters: total population/district; women of reproductive age/district; number of births/district and number of pregnancies/district.

We will compare midwives' authorisation to perform BEmONC signal functions with the country's most recent Countdown 2030 country profile and response to the most recent WHO RMNCH Policy Survey. We will then compare the tasks that midwives and midwifery professionals are authorised to perform to their reported actual performance of those tasks over the last 90-day period in facilities, where emergency maternal and newborn care are available in each study setting. We will report any variance between midwifery professionals' authorisation, training, and practice patterns.

## Indicators 6 and 7: triangulating measures of availability–validating indicators for monitoring 'Availability of B/CEmoNC facilities'

### Aim

To explore two dimensions of availability of B/CEmONC facilities: availability of all B/CEmONC signal functions within designated B/CEmONC facilities, and sufficient number of B/CEmONC facilities to meet the needs of the population (coverage). The aim is to compare the value of estimates emphasising different dimensions of availability of B/CEmONC facilities, based on different measurement approaches and data sources, to explore their external consistency or convergent validity.

### Methods

We will review records at all participating facilities where births take place to look for evidence that they have performed emergency signal functions within the previous 90 days and offer services 24 hours per day/7 days/week. We will perform geospatial analysis to estimate the travel time to each facility within the sample for various segments of the population. We will use a publicly available global population model for these estimations.

We will compare and report any variance between B/CEmONC designation and functionality across all facilities. We will calculate and compare the value of the indicator in each study district using the following denominators: 500 000 population/district; 20 000 births/district; 30 000 pregnancies/district. Last, we will use the travel time estimates obtained from the geospatial analysis to ascertain the number of facilities that are within a 2-hour travel time for the total population, for women of reproductive age, and for the number of births and pregnancies occurring to women within each study district. We will explore how the value of the indicator differs based on the denominator used and compare the values of the indicator reflecting these various approaches to measuring EmONC availability and report differences.

## Indicator 8: validating 'maternal death review coverage' to improve maternal mortality data

### Aim

To validate both numerator and denominator of the indicator 'Maternal death review coverage', defined as the percentage of maternal deaths occurring in a facility that were audited, in the study settings. Both numerator and denominator are subjected to threats to validity due to under-reporting and misclassification of maternal deaths.

### Methods

We will collect documentary evidence of maternal death and maternal death reviews in all facilities through chart and record review. We will perform retrospective review of secondary data obtained from district HMIS on both maternal deaths and maternal death reviews reported from all facilities.

We will compare the number of facility-based maternal deaths reported through HMIS to the district to the verified number of maternal deaths in all facilities in the district in patient registers. We will trace individual deaths by dates and other reported details to verify they have been reported to the district. Once validated, we will aggregate all maternal deaths reported for comparison. We will review facility death review committee records for the last 1-year period to extract the number of maternal death reviews conducted and the content of each review. We will compare the number of maternal death reviews reported to each district with the number of reviews validated through facility record review that met the definitional standard for quality[18] in the same district. Finally, we will tabulate maternal death review coverage using primary data for the numerator and denominator to the official value reported in the indicator in each country.

## Indicator 9: validating 'demand for family planning satisfied' from a woman-centered perspective: does the indicator reflect women's lived experience?

### Aim

'Demand for family planning satisfied through modern methods of contraception' uses a macroeconomic lens to look at contraceptive supply and demand, aggregating data from individual women; however, it is uncertain how well it correlates with women's own subjective perceptions of their personal demand for contraception through modern methods or how well that demand has been satisfied. This study has two aims: (1) at the individual level, to assess whether women's self-reported demand for family planning and its satisfaction converges with the standard DHS-derived measure and (2) at the population level, to examine how the value of the indicator changes based on the use of derived data from the standard calculation versus self-reported data reflecting women's own perceptions.

### Methods

We will administer a community-based survey to a sample of women in each study setting that includes direct questions to women about their desire for and use of contraception, their satisfaction with their current method and their experience of care during their most recent family planning encounter. We will then ask all the questions, in order, in the DHS algorithm used as the global standard to calculate the indicator.

We will compare the results for individual women of two different approaches to measuring the construct of 'demand for family planning satisfied through modern methods of contraception' using matched t tests. We will disaggregate by women's characteristics to identify patterns. Finally, at the population level, we will compare the value of the indicator calculated from primary data we collect to the aggregate district/province level

data reported through DHS where available to explore convergence.

## Indicator 10: comparative analysis of two scoring approaches to SDG 5.6.2. and their impact on the indicator value and interpretation of the results

### Aim

Sustainable Development Goal 5.6.2. tracks the 'Number of countries with laws and regulations that guarantee full and equal access to women and men aged 15 years and older to sexual and reproductive healthcare, information and education'. Weaknesses with the indicator scoring methodology have the potential to change its value and affect its interpretation. The aim of this exercise is to verify the laws and regulations reported for this indicator in Ghana and to explore whether the value of the indicator changes using new estimation methods to calculate its score compared with the established method, to improve interpretation.

### Methods

We will conduct a comprehensive desk review of legal statutes and regulations related to the 13 components in the indicator metadata. We will conduct secondary analysis of results from the UN 12th Inquiry Among Governments on Population and Development, Module II (Fertility, Family Planning and Reproductive Health) Survey,[19] which reports on existing laws along with barriers and enablers.

We will compare the laws and regulations on record in Ghana to the 13 components reported in the indicator for completeness and accuracy. We will calculate scores for the data collected from the UN Module II survey using the original UN scoring method and alternative scoring methods to look for differences in resulting values of the indicator. Values will be compared and sensitivity analyses conducted to explore the range of variation in the value of the indicator and the associated impact on its interpretation as a measure of sexual and reproductive health and rights.

## DISCUSSION

Because indicators for tracking maternal health system performance and effectiveness of maternal health policies rarely undergo systematic validation, methods for assessing such indicators are not codified. This research is expected to contribute new knowledge on validation methodology to the field of maternal health measurement.

Improving maternal health metrics, data quality and measurement capacity is one of the 11 Key Themes highlighted in the EPMM Strategies. The results of this research will allow data custodians to strengthen core measures for monitoring a number of critical distal determinants of maternal mortality that comprise an enabling environment for maternal health and survival.

There are some limitations to the methodology proposed in this research protocol. We expect there will be data limitations. First, data will not be national. The scope of this research study is subnational, limited to four districts in two states within each of the three research country settings. Similarly, while a census of eligible health workers of various cadres is required to answer some of the validation questions to be explored in this research, we cannot oblige all members of the study population within the research settings to consent to participate in the study; we will attempt to address such limitations to the data we collect in the analysis.

## ETHICS AND DISSEMINATION
### Ethical and safety considerations

All research partner organisations received approval to conduct human subjects research from each of their respective Institutional Review Boards (IRBs) and obtained approvals or permissions as needed from their respective Ministry of Health and other required institutions.

Research staff in each country will obtain informed consent from participants prior to data collection. All potential participants in the study will be fully informed about the objectives, their right to refuse or to withdraw and existing procedures for ensuring confidentiality. For participants below the age of majority (India and Ghana: 15–17-year old; Argentina: 15-year old) (indicators 10, 22), written consent will be obtained from the parent or legal guardian of the minor, then written assent will be obtained from the minor. Both parties must consent to participate. Documentation of consent will occur after trained research staff have described the study and answered all outstanding questions. The participant and researcher will both sign and date the consent form. Participants who are illiterate will sign the form using their thumbprint. In the case of a self-administered electronic survey, consent may be obtained electronically from the participant prior to distribution of the electronic survey.

A data security plan is registered with the IRB of the Harvard T.H. Chan School of Public Health.

This study received approval from the following Ethical Review Boards:

USA: The IRB of the Harvard T.H. Chan School of Public Health, IRB 19-1086.

Argentina:
- ► La Secretaria de Coordinación General del Sistema de Salud-Dirección Provincial del Capital Humano-Comité Provincial de Ética de la Provincia de Jujuy.
- ► El Ministerio de Salud Pública de Salta-Dirección de Recursos Humanos-Programa de docencia e investigación-Comisión provincial de investigaciones biomédicas-Comité de Ética de Investigación.
- ► El Consejo de Bioética de la Provincia de La Pampa.
- ► El Comité de Ética Central de la Provincia de Buenos Aires.

Ghana: Ghana Health Service Ethics Review Committee, GHS-ERC022/08/19.

India: Sigma IRB, 10052/IRB/19-20.

## Dissemination plan

Publication of the findings is planned through a special Collection in the PLoS Medicine journal.

Data deposition will be in the Harvard Dataverse data repository per the Bill & Melinda Gates Foundation Open Access Policy.

## Author affiliations

[1]Global Health & Population, Harvard University T H Chan School of Public Health, Boston, Massachusetts, USA

[2]Women and Health Initiative; Department of Global Health and Population, Harvard University T H Chan School of Public Health, Boston, Massachusetts, USA

[3]Department of Population, Family, and Reproductive Health, University of Ghana School of Public Health, Accra, Ghana

[4]Department of Epidemiology and Disease Control, University of Ghana School of Public Health, Accra, Greater Accra, Ghana

[5]Institute for Clinical Effectiveness and Health Policy, Buenos Aires, Argentina

[6]Population Council, New Delhi, India

[7]Department of Epidemiology and Disease Control, University of Ghana, Legon, Greater Accra, Ghana

[8]Department of Global Health and Population, Harvard University, Boston, Massachusetts, USA

**Acknowledgements** Lenka Benova, Eduardo Bergel, Ann-Beth Moller, Allisyn Moran, Jeff Blossom.

**Contributors** RRJ and JG drafted this paper with inputs from all co-authors. RJJ developed the proposal for funding with inputs from Ana Langer and colleagues from the EPMM Working Group and the Harvard T.H. Chan School of Public Health. All co-authors collaborated to conceptualise and co-develop the research aims and methods. JG led the development of the analytic plans with review and input from all co-authors. All co-authors provided substantive review feedback to finalise the paper.

**Funding** This work was supported by the Bill and Melinda Gates Foundation with grant number OPP1169546. Under the grant conditions of the Foundation, a Creative Commons Attribution 4.0 Generic License has already been assigned to the Author Accepted Manuscript version that might arise from this submission.

**Competing interests** None declared.

**Patient and public involvement** Patients and/or the public were not involved in the design, or conduct, or reporting, or dissemination plans of this research.

**Patient consent for publication** Consent obtained directly from patient(s).

**Provenance and peer review** Not commissioned; externally peer reviewed.

## ORCID iDs

R Rima Jolivet http://orcid.org/0000-0002-1440-4722

Jewel Gausman http://orcid.org/0000-0002-9880-2591

Paula Vázquez http://orcid.org/0000-0003-0206-6452

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
