## [Reviewer comments · BMJ Open]

ARTICLE DETAILS

TITLE (PROVISIONAL)	A multisite, mixed methods study to validate ten maternal health system and policy indicators in Argentina, Ghana, and India: a research protocol
AUTHORS	Jolivet, R. Rima; Gausman, Jewel; Adanu, Richard; Bando, Delia; Belizan, M; Bergel, Eduardo; Berrueta, Mabel; Chakraborty, Suchandrima; Kenu, Ernest; Khan, Nizamuddin; Odikro, Magdalene; Pingray, Veronica; Ramesh, Sowmya; Saggurti, Niranjana; Vázquez, Paula; Langer, Ana

VERSION 1 – REVIEW

REVIEWER	Azim, Tariq John Snow Inc Washington DC, International Division
REVIEW RETURNED	03-Sep-2021

GENERAL COMMENTS	Success of health program is not only measured by what has been achieved, but also on how sustainable that success is and that it will not collapse once external assistance is gone. This requires strengthening the enabling environment within the country for that program. This applies to any health program, and more so to maternal and newborn health. The authors have picked up a very important area that can contribute to that end. As the saying goes "what gets measured, gets managed". Thus, measuring the progress made in building a sustainable enabling environment for ending preventable maternal mortality is a necessity to build and sustain that enabling environment. Obviously, measurement of the components of that enabling environment is a complex construct. The indicators that were developed for that purpose are novel and hence require to pass the test of validity. The authors have proposed a very systematic and thorough methodology/protocol to test that validity of selected ten indicators. They have elaborately presented the purpose and methodology to do so and these are based on already tested and published procedures. When I started to review this protocol, I was apprehensive of authors proposing two novel approaches, i.e., a novel methodology of validating complex indicators and then using that novel method to test novel indicators. After going through the protocol, I am convinced that the methodology to be used is based on established procedures. Publishing this protocol will also serve as an useful resource to inform similar research in maternal health as well as other aspects of global health. The authors have done well to describe the statistical methods to be used for analysis. However, the limitations of the study are not discussed in the main body of the text. Having a section to describe the study limitations will make this protocol more acceptable to the wider audience.
--

REVIEWER	Marchant, Tanya London School of Hygiene and Tropical Medicine, Public Health and Policy
REVIEW RETURNED	13-Sep-2021

GENERAL COMMENTS	Thank you for the opportunity to review this excellent article. The work described in the protocol is highly innovative, describing research methods to validate system and policy-level indicators. The detail and transparency around methods will advance the maternal health measurement field. Text is written to a very high quality and is comprehensive. I have a few suggestions that might help improve clarity, for consideration. These comments are also elaborated upon in the marked-up article file. Introduction The flow of information might be edited so that the reader is told about the need to develop and validate system/policy-level indicators prior to the explanation about types of validation and current evidence base on indicator validity. Methods and analysis There are lots of concepts for the reader to understand. Of central importance, early on the reader should grasp the 10 indicators that will be validated through this research. At the moment two styles limit this. First, I found Figure 1 difficult to understand – the link between the content of the 11 boxes, the 10 indicators being validated, and the list of 11 EPMM themes in the figure is not immediately obvious and either needs more explanation or a different presentation. Perhaps the figure aims to justify why 10 indicators are defined for 11 themes but could do so more directly? Second, consider dropping the use of '7 validation exercises' throughout, replacing this with indicator numbering (1-10): some indicators went through the same process which is simple to state. The current format suggests emphasis on the 7 exercises but I thought the emphasis should be on the 10 indicators being validated. A short paragraph could be added to close the article, perhaps reinforcing which new evidence will be generated through this research and what will change as a result.
---

VERSION 1 – AUTHOR RESPONSE

Reviewer: 1

We have added a short Conclusions section where we reprise the strengths and limitations, as well as the new evidence to be generated by this research and expected implications, in response to similar requests from both Reviewers.

Reviewer: 2

In general, we have incorporated all suggested edits in the marked-up article file.

We have implemented the suggestion to move the text describing the need for indicator validation above the literature review and the explanation about types of validation methods.

We appreciate the comment about Figure 1. and the opportunity to make the intent clearer for readers. We believe that Figure 1 illustrates an important point, which we have attempted to articulate more clearly, as it states the overall research aim and situates it within the context of efforts to strengthen and advance the EPMM Strategies report, the global strategic framework for maternal health.

We have revised the text in the body of the paper that precedes and explains Figure 1., which now reads, “The overall aim of the study is to improve maternal health measurement by validating ten indicators from the EPMM monitoring framework, in order to drive improvement and track progress towards achieving the priority recommendations outlined in the EPMM Strategies. Of note, this research assesses 40% of the indicators in the set of EPMM metrics designed to allow countries and global partners to monitor critical dimensions of the upstream enabling environment for maternal health. Furthermore, the indicators validated through this research reflect a broad range of these distal determinants, as they correspond to nine out of the 11 EPMM Key Themes (Figure 1).”

Moreover, we have revised the Figure as follows:

- We have simplified the title to read “Indicators for Validation by EPMM Key Theme”
- We have added a legend with reference citation that reads, “The ten indicators selected for validation through this study (in the outer boxes) were each ranked as one the strongest available indicators for monitoring the corresponding EPMM Key Theme (listed in the box in the center). (5)”
- For the two EPMM Key Themes for which no indicator will be validated through this study, we have clarified the text to read, “No EPMM indicator selected for validation through this study.”

We hope these revisions meet with your satisfaction and resolve the confusion.

Thank you for sharing the comment regarding “7 validation exercises”. We have implemented your suggestion to put the emphasis on the 10 indicators to be validated through the research. Two pairs of indicators will be cross-validated through a shared specific study design and methodology to answer one set of validation question that will allow us to explore their convergence and relative validity based on evidence; for this reason, we have retained the structure of 7 discrete studies within the overall research plan.

To respond to your feedback, we have revised Table 2 as follows:

- We have removed Column 1 which listed 7 numbered Validation Exercises.
- Instead we have numbered the indicators 1- 10.
- For indicators that will be validated in tandem, we have added an asterisk with the following legend to the box: “(*The validity of these two indicators designed to measure a related construct will be evaluated in tandem using the same research validation questions.)”

Similarly, in the body of the paper the subheadings for each validation study have been changed to emphasize the indicators they address.

We have added a short Conclusions section where we reprise the strengths and limitations, as well as the new evidence to be generated by this research and expected implications, in response to similar requests from both Reviewers.

VERSION 2 – REVIEW

REVIEWER	Azim, Tariq John Snow Inc Washington DC, International Division
REVIEW RETURNED	28-Oct-2021

GENERAL COMMENTS	I have a few more comments in the attached manuscript. Addressing those will further improve the clarity of the paper in terms of its study design and definitions.
---

REVIEWER	Marchant, Tanya London School of Hygiene and Tropical Medicine, Public Health and Policy
REVIEW RETURNED	11-Oct-2021

GENERAL COMMENTS	Thank you for the clear explanation of changes made to this excellent manuscript
--

VERSION 2 – AUTHOR RESPONSE

Reviewer: 1

Dr. Tariq Azim, John Snow Inc Washington DC, The George Washington University Milken Institute of Public Health

Comments to the Author:

I have a few more comments in the attached manuscript. Addressing those will further improve the clarity of the paper in terms of its study design and definitions.

1. “Couldn't find any mention of the Box 1 in the text. How is this information in the box relevant to the study? (It is, but need a line in the text to make this information in the box relevant)”.
2. Box 1: Is this proposed by the authors or can be referenced to any publication?
3. Page 7, Lines 44-48: Suggestion to insert reference for Box 1 here
4. Page 8, Line 7: Is this study that framework published by Benova? Is that the one in Box 1?

In response to the multiple comments asking for clarification about Box 1, the provenance of the information therein, and its relevance to the study, we have made the following changes:

We have added a reference to Box 1 as well as an explanation of the relevance of this information, which come from the Benova 2020 study, in the paragraph that cites that study. It now reads: “Benova et al. (14) (2020) published a conceptual framework compiling definitions of indicator validity (Box 1) and approaches for assessing its various dimensions, based on interviews with practitioners of MNH measurement. The framework includes methodological approaches for assessing validity of indicators for tracking health policy and health system factors, and calls for more research in this domain. We used this framework to inform the development of our research methods, based on specific validation questions of relevance to each indicator undergoing assessment.”

The citations for references used to populate Box 1 have now been added.

5. Page 8, line 28: Maybe in this methods section, authors can mention how validity is assessed, i.e., what metrics or framework

We thank the reviewer for this comment. Because there is no standard approach (metric or framework) for assessing construct or convergent validity, we have developed a tailored analytical plan with appropriate statistics to compare the values of the reported indicators to evidence collected in each case.

We have amended the description of the methods to read as follows: " This observational study explores the validity of ten indicators from the EPMM monitoring framework. It utilizes mixed methods, including 1) cross-sectional review of secondary policy, legal, and regulatory data, 2) retrospective review of facility-level patient and administrative data, and 3) collection of primary, quantitative, cross-sectional data from health service providers and clients. There isStandard approaches for assessing the validity of policy and health system indicators are not available; therefore, we developed a specific methodological approach to validate each indicator, tailored to test the validation questions that reflect the specific aims and research questions relevant to each indicator undergoing validation and its underlying construct. Because there is no standard approach (metric or framework) for assessing validity of indicators of upstream health system functionality, we have developed a tailored analytical plan with appropriate statistics to compare the values of the reported indicators to evidence collected in each case. In two specific cases, two indicators designed to monitor a similar construct are compared to each other to explore their convergence and whether indicator adjustment could improve measure validity for that construct. These two indicator pairs share the same validation research questions and are studied in tandem. Thus, the validity of the ten EPMM indicators is evaluated via seven separate assessments, or validation exercises."

The specific methods and analytic plan for each indicator are described in the body of the protocol.

6. Table 2: It would be good to provide clue (maybe in parenthesis) to whether each questions is going to help assess content, criterion or construct validity. This will help to establish the scope of the study in terms of how each indicator's validity was established.

Thank you for this suggestion, which we have now implemented.

Reviewer: 1

Competing interests of Reviewer: No competing interests

Reviewer: 2

Dr. Tanya Marchant, London School of Hygiene and Tropical Medicine

Comments to the Author:

Thank you for the clear explanation of changes made to this excellent manuscript

Reviewer: 2

Competing interests of Reviewer: None